# THINKING FOURTH DIMENSIONALLY:
# TREATING TIME AS A RANDOM VARIABLE IN EBMS

## ABSTRACT

Recent years have seen significant progress in techniques for learning high-dimensional distributions. Many modern methods, from diffusion models to Energy-Based-Models (EBMs), adopt a coarse-to-fine approach. This is often done by introducing a series of auxiliary distributions that gradually change from the data distribution to some simple distribution (*e.g.,* white Gaussian noise). Methods in this category separately learn each auxiliary distribution (or transition between pairs of consecutive distributions) and then use the learned models sequentially to generate samples. In this paper, we offer a simple way to generalize this idea by treating the "time" index of the series as a random variable and framing the problem as that of learning a single joint distribution of "time" and samples. We show that this joint distribution can be learned using any existing EBM method and that it allows achieving improved results. As an example, we demonstrate this approach using contrastive divergence (CD) in its most basic form. On CIFAR-10 and CelebA ($32 \times 32$), this method outperforms previous CD-based methods in terms of inception and FID scores.

## 1 INTRODUCTION

Probability density estimation is among the most fundamental tasks in unsupervised learning. It is used in a wide array of applications, from image restoration and manipulation (Nichol et al., 2021; Du et al., 2021; Lugmayr et al., 2020; Kawar et al., 2021; 2022) to out-of-distribution detection (Du & Mordatch, 2019; Grathwohl et al., 2019; Zisselman & Tamar, 2020). However, directly fitting an explicit probability model to high-dimensional data is a hard task, particularly when the data samples concentrate around a low-dimensional manifold, as is often the case with visual data. One way to circumvent this obstacle is by using coarse-to-fine approaches. In fact, in one form or another, coarse-to-fine strategies have been used with great success in most types of generative models (both implicit and explicit), including generative adversarial networks (GANs) (Karras et al., 2018), variational autoencoders (VAEs) (Vahdat & Kautz, 2020), energy-based models (EMBs) (Gao et al., 2018; Zhao et al., 2020), score matching (Song & Ermon, 2019; Li et al., 2019) and diffusion models (Sohl-Dickstein et al., 2015; Ho et al., 2020).

The coarse-to-fine idea is commonly implemented through the introduction of a series of auxiliary distributions that gradually transition from the data distribution to some simple known distribution that is smoothly spread in space (*e.g.,* a standard normal distribution). This construction is illustrated in Fig. 1a for a two-dimensional data. The index running over the series of distributions is typically referred to as "time". This is to reflect either the diffusion-like sequential manner in which samples are generated for training (from fine to coarse) (Sohl-Dickstein et al., 2015; Ho et al., 2020) or the annealing-like sequential order in which samples are generated from the model at test time (from coarse to fine) (Song & Ermon, 2019). Methods that use this construction attempt to learn each of the distributions in the series (or each transition rule between pairs of consecutive distributions) *separately of the other distributions*[1]

In this paper, we explore a more general approach for exploiting the coarse-to-fine structure, which can be used in conjunction with almost any explicit distribution learning algorithm and leads to

---

[1]It is common to represent all models by a single neural network that accepts the "time" index as input. But for each "time" step, the network is exposed only to samples from the corresponding distribution.

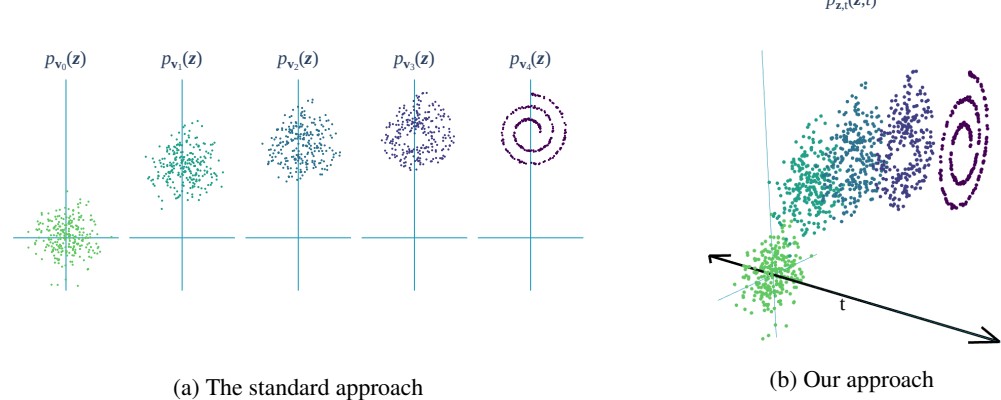

(a) The standard approach

(b) Our approach

Figure 1: (a) Coarse-to-fine distribution learning methods introduce a series of auxiliary distributions that gradually transition from the data distribution (2D spiral in this example) to some simple distribution (a Gaussian here). These methods learn each auxiliary distribution (or pair of consecutive distributions) separately. (b) Here we treat the "time" index of the series as a random variable, t, and the samples from all distributions as samples from a single random vector $\mathbf{z}$. We then train the model to learn the joint distribution $p_{\mathbf{z},t}(t, \mathbf{z})$ using samples $(t, \mathbf{z})$.

improved results. The key idea is to gather the samples from all auxiliary distributions and view them as coming from a single joint distribution. More specifically, we treat the "time" index of the series as a random variable, t, and the samples from all auxiliary distributions as samples of a random vector $\mathbf{z}$. This allows learning a single model for the joint distribution $p_{\mathbf{z},t}(\mathbf{z}, t)$, using pairs of samples $(\mathbf{z}, t)$ (see Fig. 1b).

To understand the benefit of this joint modeling, it is important to note that many of the individual distributions $p_{\mathbf{z}|t}(\mathbf{z}|t)$ commonly occupy only small regions of the space. Thus, when training a separate model for each $t$, each model is accurate over a different region in space, which can lead to inaccuracies at test time when switching between models. In contrast, here we learn the joint distribution $p_{\mathbf{z},t}(\mathbf{z}, t)$, either directly (Sec. 3.4) or by breaking the problem in a reverse way and learning $p_{\mathbf{z}}(\mathbf{z})$ and $p_{t|\mathbf{z}}(t|\mathbf{z})$ (Sec. 3.3). Thus, during training, our unified model is exposed to samples from the entire space, leading to better stitching of the different parts.

Once a model is trained using our approach, it can be used similarly to existing methods by extracting the auxiliary distributions $p_{\mathbf{z}|t}(\mathbf{z}|t)$ and sampling from them one after the other, from coarse to fine. It can also be used in alternative ways, as we discuss in Sec. 3.5.

To illustrate the strength of our approach, we apply it together with the *vanilla contrastive divergence (CD)* method (Hinton, 2002) on the CIFAR10 (Krizhevsky et al., 2009) and CelebA (Liu et al., 2015) ($32 \times 32$) datasets. It is important to note that although the vanilla CD method is theoretically justified (Yair & Michaeli, 2020), it fails when directly applied to high dimensional visual data (Gao et al., 2018). This is because it provides good estimates only nearby the data manifold. To date, good results have been obtained only with persistent contrastive divergence (PCD) (Tieleman, 2008; Du & Mordatch, 2019), which maintains a buffer of past samples. With our approach, on the other hand, plain CD not only succeeds in learning the distribution, but it also improves upon all previous PCD-based techniques in terms of Inception Score (IS) and Fréchet Inception Distance (FID).

## 2    RELATED WORK

The idea of learning an explicit generative model by using an auxiliary coarse-to-fine series of distributions, has been used in many works. We briefly mention its use within popular models.

Song & Ermon (2019) constructed a series of distributions by adding increasing amounts of white Gaussian noise to the training samples. They learned the gradients of the distributions using denoising score matching (Vincent, 2011), and used the trained model to solve various generative tasks using gradient based simulated annealing.

Diffusion models (Sohl-Dickstein et al., 2015; Ho et al., 2020; Song et al., 2020; Dhariwal & Nichol, 2021) are currently the state-of-the-art methods for image generation. In this approach, the series of distributions is defined through a diffusion process that begins with samples from the training set and gradually transforms the distribution into white Gaussian noise. A series of models (denoisers) is then trained to capture the reverse process, allowing to generate samples from each distribution given the preceding one. This process can also be viewed as starting from a noisy image and repeatedly removing small portions of the noise until reaching a clean image (Ho et al., 2020).

Gao et al. (2020b) used a construction similar to that of diffusion models, but with much fewer distributions. They employed a conditional version of maximum likelihood for training an EBM for each distribution in the series. This was done using short MCMC chains for generating adversarial samples from a selected distribution based on samples from consecutive distributions. The adversarial samples are then used along with true samples from the selected distribution for calculating the optimization objective.

Rhodes et al. (2020) address an arbitrary series of distributions that transitions between the dataset and some reference distribution. A series of classifiers is then trained to discriminate between each pair of consecutive distributions. The fact that the ratio between the distributions can be extracted from these classifiers is then used for computing the target distribution in a telescopic manner.

## 3 METHOD

### 3.1 INTRODUCING THE JOINT DISTRIBUTION

Given a dataset of i.i.d. samples $\{\boldsymbol{x}_i\}$, we would like to learn a model for the distribution $p_{\mathbf{x}}$ from which the samples were drawn. To do so, we first select some known distribution $p_{\mathbf{n}}$ as a reference. A possible choice can be white Gaussian noise. We then introduce a series of $T$ auxiliary distributions $\{p_{\mathbf{v}_t}\}_{t=0}^{T-1}$ which begin with the reference distribution, $p_{\mathbf{v}_0} = p_{\mathbf{n}}$, and end with the distribution of the data, $p_{\mathbf{v}_{T-1}} = p_{\mathbf{x}}$. The main requirements from those distributions are that we know how to draw samples from them and that the effective overlap between consecutive distributions be small. One common construction relies on linear combinations of samples from $p_{\mathbf{x}}$ and $p_{\mathbf{n}}$. That is, given a sample from the dataset, $\boldsymbol{x}$, and a sample from the reference distribution, $\boldsymbol{n}$, an intermediate sample $\boldsymbol{v}_t$ can be generated as

$$\boldsymbol{v}_t = \alpha_t \boldsymbol{x} + \beta_t \boldsymbol{n} \tag{1}$$

with some predefined coefficients $\{\alpha_t, \beta_t\}_{t=0}^{T-1}$ that monotonically transition from $(\alpha_0, \beta_0) = (0, 1)$ to $(\alpha_{T-1}, \beta_{T-1}) = (1, 0)$.

Up to this point, this is the standard coarse-to-fine construction. Here, however, we proceed to view the samples from all distributions $\{p_{\mathbf{v}_t}\}_{t=0}^{T-1}$ as coming from a single joint distribution. Specifically, we introduce two random variables. The first is a random "time" index $\mathrm{t} \sim p_{\mathrm{t}}$, which is used to select an auxiliary distribution from the series. A simple choice would be a uniform distribution over $[0, T-1]$ (in Sec. 3.5 we discuss how $p_{\mathrm{t}}$ can be modified at test time to aid the sample generation process). Using t, we define a second random variable, $\mathbf{z} = \boldsymbol{v}_{\mathrm{t}}$. Namely, $\mathbf{z}$ is a random draw from one of the auxiliary distributions, where the index is randomly chosen according to t, so that

$$p_{\mathbf{z}|\mathrm{t}}(\boldsymbol{z}|t) = p_{\mathbf{v}_t}(\boldsymbol{z}). \tag{2}$$

Given these two variables, we define our auxiliary problem as that of learning the joint distribution $p_{\mathbf{z},\mathrm{t}}$ based on samples of $(\boldsymbol{z}, t)$.

Learning the joint distribution can be done in several ways:

1. Using the decomposition $p_{\mathbf{z},\mathrm{t}}(\boldsymbol{z}, t) = p_{\mathbf{z}|\mathrm{t}}(\boldsymbol{z}|t)p_{\mathrm{t}}(t)$. Since $p_{\mathrm{t}}$ is known, this only requires learning $p_{\mathbf{z}|\mathrm{t}}$, which is the standard approach[2]. Particularly, for each $t \in \{0, \ldots, T-1\}$, the distribution $p_{\mathbf{z}|\mathrm{t}}(\cdot|t)$ is learned using an optimization problem that is defined only in terms of samples corresponding to that $t$. This approach is illustrated in Fig. 1a.

---

[2]Strictly speaking, when using EBM-based methods, each $p_{\mathbf{z}|\mathrm{t}}$ is learned up to an unknown normalization constant, preventing computation of $p_{\mathbf{z},\mathrm{t}}$. Methods using this technique, only use the individual $p_{\mathbf{z}|\mathrm{t}}$ at test time.

2. Using the decomposition $p_{\mathbf{z},t}(\mathbf{z},t) = p_{t|\mathbf{z}}(t|\mathbf{z})p_{\mathbf{z}}(\mathbf{z})$. Here we need to learn both terms, which are illustrated in Figs. 2b and 2a.

   (a) Learning $p_{t|\mathbf{z}}$ amounts to solving a classification problem (*i.e.,* learning to predict the discrete "time" variable $t \in \{0, \ldots, T-1\}$ from a sample $\mathbf{z}$). This can be done using the cross-entropy loss.

   (b) The distribution $p_{\mathbf{z}}$ can be learned *e.g.,* using any EBM learning method. Importantly, this is a significantly simpler task than learning $p_{\mathbf{z}|t}$ (especially for the finer $t$'s), as $p_{\mathbf{z}}$ is a smoother function that is more spread in space.

3. Directly learning $p_{\mathbf{z},t}$, which appears in Fig. 1b. This requires slightly adapting existing methods as this is a joint distribution over two domains with different properties.

It should be noted that although each of these options suffices on its own for extracting $p_{\mathbf{z},t}$, it is also possible to use combinations of these losses in order to further improve the training (*e.g.,* a loss on $p_{t|\mathbf{z}}$ in addition to the standard loss on $p_{\mathbf{z}|t}$).

Our key observation is that substantially improved results are obtained when using the losses of Option 2 and/or 3 in conjunction or instead of the standard loss on $p_{\mathbf{z}|t}$ (Option 1). This is because the losses on $p_{t|\mathbf{z}}$, $p_{\mathbf{z}}$, and $p_{\mathbf{z},t}$ are optimized using samples from all auxiliary distributions. This is while the standard approach of training a model per $t$ using a loss on $p_{\mathbf{z}|t}$, involves samples only from that particular $t$ and these samples come from a very restricted region in space.

In Sections 3.3 and 3.4 we show how options 2 and 3, respectively, can be used in conjunction with the CD method. Option 2 leads to better results than Option 3, and improves upon all previous CD-based methods.

## 3.2 THE PARAMETRIC MODEL

As opposed to the common approach in which $t$ is fed as input to the model (Song & Ermon, 2019; Rhodes et al., 2020; Ho et al., 2020), here we propose to have the model output a vector of length $T$ that contains the values of $\log p_{\mathbf{z},t}(\mathbf{z},t)$ for all $t \in \{1, \ldots, T-1\}$. Namely, we use a parametric model $\boldsymbol{f_\theta}$ (a neural network in our experiments), which accepts $\mathbf{z}$ as input and outputs

$$\boldsymbol{f_\theta}(\mathbf{z}) = \left[\log \hat{p}_{\mathbf{z},t;\theta}(\mathbf{z}, t=0), \log \hat{p}_{\mathbf{z},t;\theta}(\mathbf{z}, t=1), \ldots, \log \hat{p}_{\mathbf{z},t;\theta}(\mathbf{z}, t=T-1)\right]^\top. \quad (3)$$

Here $\hat{p}$ denotes the model's estimate of the probability density $p$. This is illustrated in Fig. 3.

In this design, each element of the output vector is basically an EBM for the corresponding auxiliary distribution (here we define the log probabilities without the minus sign). This is related to the observation in (Grathwohl et al., 2019), which draws the connection between classifiers and EBMs. However, as opposed to EBMs, here we strive to learn a normalized distribution. This can be done

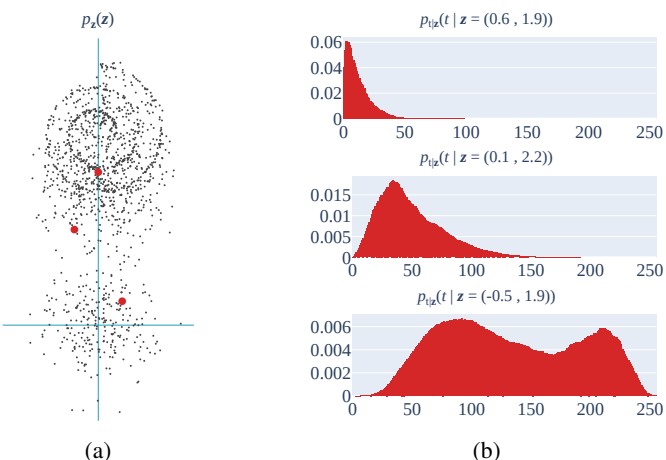

|     |     |
| :-: | :-: |
| (a) | (b) |

Figure 2: (a) Samples from the marginal distribution $p_{\mathbf{z}}$. (b) The conditional distribution $p_{t|\mathbf{z}}$ for the three red points of $\mathbf{z}$ in (a).

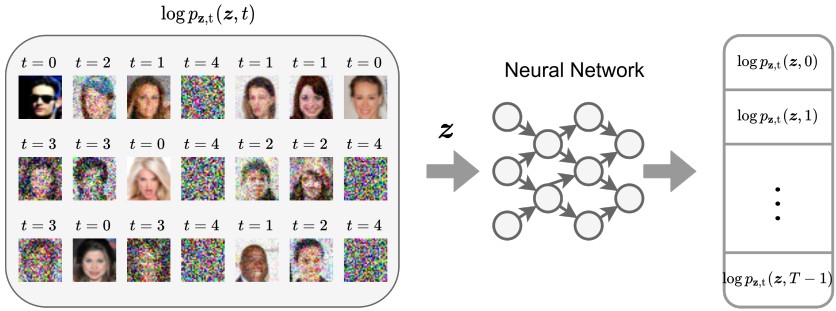

Figure 3: Our parametric model consumes a sample $z$ and outputs a vector of log probabilities indicating the joint likelihoods of $z$ and $t$ for all values of $t = 0, \ldots, T-1$.

by using an optimization process that mixes values along with using the fact that $p_{\mathbf{z}|\mathbf{t}}(z|0) = p_{\mathbf{n}}(z)$ in order to fix the first element of the output vector of $\boldsymbol{f_\theta}(z)$ to be $\log p_{\mathbf{n}}(z)$.

Defining the model this way allows efficiently computing $\log \hat{p}_{\mathbf{z};\boldsymbol{\theta}}(z)$ and $\log \hat{p}_{\mathbf{t}|\mathbf{z};\boldsymbol{\theta}}(t|z)$, as

$$\log \hat{p}_{\mathbf{z};\boldsymbol{\theta}}(z) = \text{logSumExp}\left(\boldsymbol{f_\theta}(z)\right), \tag{4}$$

$$\log \hat{p}_{\mathbf{t}|\mathbf{z};\boldsymbol{\theta}}(t|z) = \text{logSoftmax}\left(\boldsymbol{f_\theta}(z)\right)_t. \tag{5}$$

These terms can be used both during training (for applying losses on these distributions) and, as we discuss in Sec. 3.5, at test time (for generating samples from the model).

In order to aid the training of the model, we make use of an observation made by Rhodes et al. (2020), stating that it is beneficial to represent a ratio of distributions as a product of many small ratio terms. We apply this idea to our model by adding an additional cumulative-summation layer $(g(\boldsymbol{x})_i = \sum_{j=0}^{i} x_j)$ before the output of the network. This additional layer does not impact the representation power of the network as it is an invertible linear operation that can, in theory, be absorbed into the preceding linear layer. However, this layer does affect the optimization of the model, as it modifies the initialization of the effective linear operation preceding the output, as well as the optimization dynamics. In practice, we found this implicit bias to be crucial for the success of the model when applied to high-dimensional distributions.

## 3.3 TRAINING THE MODEL USING CD+CE

In this section we describe how our suggested model can be trained using a CD loss on the marginal distribution $p_{\mathbf{z}}$ together with a cross-entropy (CE) loss on $p_{\mathbf{t}|\mathbf{z}}$. The overall loss in this method is therefore $\mathcal{L}_{\text{CD}}(\boldsymbol{\theta}) + \mathcal{L}_{\text{CE}}(\boldsymbol{\theta})$, and we refer to it as CD+CE (see Alg. 1).

**Contrastive divergence** The CD method uses an MCMC process to generate contrastive samples at each training iteration. Specifically, let us denote by $\mathcal{T}_{\hat{p}_{\mathbf{z};\boldsymbol{\theta}}}$ the transition operator of an MCMC process (an operator that performs a single MCMC step) designed to draw samples from $\hat{p}_{\mathbf{z};\boldsymbol{\theta}}$. Note that in each training iteration, the MCMC process operates with respect to the current estimate of the joint distribution. To generate a contrastive sample, the MCMC process is initialized with a sample from the dataset, $\tilde{z}^0 = z$, and is run for $K$ steps, $\tilde{z}^{k+1} = \mathcal{T}_{\hat{p}_{\mathbf{z};\boldsymbol{\theta}}}(\tilde{z}^k)$. This results in a contrastive sample $\tilde{z} = \tilde{z}^K$. The CD loss is then defined as

$$\mathcal{L}_{\text{CD}}(\boldsymbol{\theta}) = \mathbb{E}\left[\log \hat{p}_{\mathbf{z};\boldsymbol{\theta}}(\tilde{\mathbf{z}}) - \log \hat{p}_{\mathbf{z};\boldsymbol{\theta}}(\mathbf{z})\right]$$

$$= \mathbb{E}\left[\text{logSumExp}\left(\boldsymbol{f}(\tilde{\mathbf{z}})\right) - \text{logSumExp}\left(\boldsymbol{f}(\mathbf{z})\right)\right], \tag{6}$$

where the expectation is overdraws of contrastive samples (first term) and samples from the dataset (second term). A popular choice for an MCMC process over a continuous distributions is Langevin dynamics in which the transition operator is given by:

$$\mathcal{T}_{\hat{p}_{\mathbf{z};\boldsymbol{\theta}}}\left(\tilde{z}^k\right) = \tilde{z}^k + \frac{\mu^2}{2}\nabla_{\mathbf{z}}\log \hat{p}_{\mathbf{z};\boldsymbol{\theta}}\left(\tilde{z}^k\right) + \mu\varepsilon$$

$$= \tilde{z}^k + \frac{\mu^2}{2}\nabla_{\mathbf{z}}\text{logSumExp}\left(\boldsymbol{f}(\mathbf{z})\right) + \mu\varepsilon, \tag{7}$$

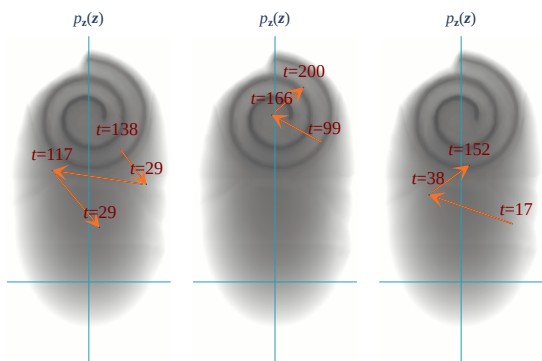

Figure 4: We present three MCMC processes running over the joint distribution of the toy model. Note that the intermediate $t$ values are used neither in CD+CE nor in JointCD, and are shown here only for completeness. Particularly, in JointCD the MCMC runs over the marginal distribution of $\mathbf{z}$, so that $t$ plays no role, and in JointCD we draw $t$ only once for the final point.

where $\boldsymbol{\varepsilon} \sim \mathcal{N}(0, \boldsymbol{I})$ and $\mu$ is the step size. To keep the MCMC process accurate, we use a Metropolis-Hastings rejection step (Hastings, 1970) as part of the transition operator.

**Cross entropy**   Learning $p_{\mathrm{t}|\mathbf{z}}$ can be achieved by minimizing the standard cross-entropy loss over the outputs of our model,

$$
\begin{aligned}
\mathcal{L}_{\mathrm{CE}}(\boldsymbol{\theta}) &= -\mathbb{E}\left[\log \hat{p}_{\mathrm{t}|\mathbf{z};\boldsymbol{\theta}}(\mathrm{t}|\mathbf{z})\right] \\
&= -\mathbb{E}\left[\operatorname{logSoftmax}\left(\boldsymbol{f}(\mathbf{z})\right)_t\right].
\end{aligned}
\tag{8}
$$

This is equivalent to training a classifier to predict the discrete "time" index $t$ given a sample $\mathbf{z}$.

### 3.4   TRAINING THE MODEL USING JOINTCD

Our suggested model can also be trained using a CD loss on the joint distribution $p_{\mathbf{z},\mathrm{t}}$. We refer to this method as JointCD (see Alg. 2). The distribution $p_{\mathbf{z},\mathrm{t}}$ is visualized in Fig. 1b.

To apply CD on $p_{\mathbf{z},\mathrm{t}}$, we need a transition operator $\mathcal{T}_{\hat{p}_{\mathbf{z},\mathrm{t};\boldsymbol{\theta}}}$ of an MCMC process that is designed to draw samples from $\hat{p}_{\mathbf{z},\mathrm{t};\boldsymbol{\theta}}$. Having such an operator, we can initialize the process with a sample from the dataset, $(\tilde{\mathbf{z}}^0, \tilde{t}^0) = (\mathbf{z}, t)$, and run $(\tilde{\mathbf{z}}^{k+1}, \tilde{t}^{k+1}) = \mathcal{T}_{\hat{p}_{\mathbf{z},\mathrm{t};\boldsymbol{\theta}}}(\tilde{\mathbf{z}}^k, \tilde{t}^k)$ for $K$ steps to generate a contrastive sample $(\tilde{\mathbf{z}}, \tilde{t}) = (\tilde{\mathbf{z}}^K, \tilde{t}^K)$. The problem is that popular MCMC techniques, like Langevin Dynamics and Hamiltonian Monte Carlo (HMC), are relevant only for continuous distributions, whereas in our case $p_{\mathbf{z},\mathrm{t}}$ is a mixed distribution ($\mathbf{z}$ is continuous and t is discrete). Nevertheless, as we show in in App. A, any continuous MCMC process can be extended to work on our joint mixed distribution simply by performing a step on $\tilde{\mathbf{z}}$ using the continuous MCMC operator $\mathcal{T}_{\hat{p}_{\mathbf{z};\boldsymbol{\theta}}}$ and then sampling $\tilde{t}$ from $\hat{p}_{\mathrm{t}|\mathbf{z}}$. Namely, a single MCMC step in our case takes the form

$$
\mathcal{T}_{\hat{p}_{\mathbf{z},\mathrm{t};\boldsymbol{\theta}}}(\tilde{\mathbf{z}}^k, \tilde{t}^k) = 
\begin{cases}
\tilde{\mathbf{z}}^{k+1} &= \mathcal{T}_{\hat{p}_{\mathbf{z};\boldsymbol{\theta}}}(\tilde{\mathbf{z}}^k) \\
\tilde{t}^{k+1} &\sim \hat{p}_{\mathrm{t}|\mathbf{z};\boldsymbol{\theta}}(\cdot|\tilde{\mathbf{z}}^{k+1}).
\end{cases}
\tag{9}
$$

We show in App. A that this process obeys the detailed balance criterion and thus its stationary distribution is indeed $\hat{p}_{\mathrm{t},\mathbf{z}}$ as desired. Note that the intermediate $\tilde{t}^k$ values are not required for sampling the next step. Therefore, in practice, we draw only the last one, $\tilde{t}^K$. We illustrate this joint MCMC process in Fig. 4.

| **Algorithm 1:** CD + CE | **Algorithm 2:** JointCD |
|---|---|
| **while** *not converged* **do** | **while** *not converged* **do** |
|     Sample $t \sim p_\mathrm{t}$, $\boldsymbol{z} \sim p_{\boldsymbol{z}\mid \mathrm{t}=t}$ |     Sample $t \sim p_\mathrm{t}$, $\boldsymbol{z} \sim p_{\boldsymbol{z}\mid \mathrm{t}=t}$ |
|     $\tilde{z} \leftarrow \boldsymbol{z}$ |     $\tilde{z} \leftarrow \boldsymbol{z}$ |
|     **for** $1$ *to* $K$ **do** |     **for** $1$ *to* $K$ **do** |
|         $\tilde{z} \leftarrow \mathcal{T}_{\hat{p}_{\boldsymbol{z};\boldsymbol{\theta}}}(\tilde{z})$ |         $\tilde{z} \leftarrow \mathcal{T}_{\hat{p}_{\boldsymbol{z};\boldsymbol{\theta}}}(\tilde{z})$ |
|     **end** |     **end** |
|   |     Sample $\tilde{t}$ from $\hat{p}_{\mathrm{t}\mid \boldsymbol{z}=\tilde{z};\boldsymbol{\theta}}$ |
|     Take gradient step on |     Take gradient step on |
|     $\log \hat{p}_{\boldsymbol{z};\boldsymbol{\theta}}(\tilde{z}) - \log \hat{p}_{\boldsymbol{z};\boldsymbol{\theta}}(\boldsymbol{z}) - \log \hat{p}_{\mathrm{t}\mid \boldsymbol{z};\boldsymbol{\theta}}(t\mid \boldsymbol{z})$, |     $\log \hat{p}_{\boldsymbol{z},\mathrm{t};\boldsymbol{\theta}}(\tilde{z},\tilde{t}) - \log \hat{p}_{\boldsymbol{z},\mathrm{t};\boldsymbol{\theta}}(\boldsymbol{z},t)$, |
|     computing the densities using (4),(5). |     computing the densities using (3). |
| **end** | **end** |

### 3.5 SAMPLING FROM THE MODEL

Sampling from the trained model can be done via simulated annealing. In this approach, one runs an MCMC process while constantly replacing the underlying distribution, starting from a simple smooth distribution and gradually refining it into the target distribution. In our context, this is commonly done by running through the series of learned auxiliary distributions, starting with $p_{\boldsymbol{z}\mid \mathrm{t}=0}$ and gradually increasing $t$ until reaching $p_{\boldsymbol{z}\mid \mathrm{t}=T-1}$ (which equals $p_{\boldsymbol{x}}$). This algorithm is outlined in Alg. 3 in App. B, and illustrated in Fig. 5a.

We note that our approach allows viewing simulated annealing as a special case of a more general sampling scheme. Specifically, we can interpret simulated annealing as running through a series of distributions $p_{\boldsymbol{z}'}^{(n)}$, where $p_{\boldsymbol{z}'\mid \mathrm{t}'}^{(n)}(\boldsymbol{z}\mid t) = p_{\boldsymbol{z}\mid \mathrm{t}}(\boldsymbol{z}\mid t)$ for all $n$ and $p_{\mathrm{t}'}^{(n)}(t) = \delta(t-n)$ (here $\delta(\cdot)$ denotes kronecker's delta function). We can therefore generalize this method by using any sequence of distributions $\{p_{\mathrm{t}'}^{(n)}(t)\}$ whose centers of mass gradually move from the small values of $t$ to the larger ones. In this generalized setting, we have $p_{\boldsymbol{z}',\mathrm{t}'}^{(n)}(\boldsymbol{z},t) = p_{\boldsymbol{z}\mid \mathrm{t}}(\boldsymbol{z}\mid t)p_{\mathrm{t}'}^{(n)}(t) = p_{\boldsymbol{z},\mathrm{t}}(\boldsymbol{z},t)p_{\mathrm{t}'}^{(n)}(t)/p_{\mathrm{t}}(t)$, from which $p_{\boldsymbol{z}'}^{(n)}$ can be extracted by summation over $t$. The resulting generalized simulated annealing algorithm is outlined in Alg. 4 in App. B. One particular choice of $p_{\mathrm{t}'}^{(n)}$ is $p_{\mathrm{t}'}^{(n)} = \mathrm{Uniform}[t(n), T-1]$ where $t(n)$ is a linear function growing from 0 to $T-1$. For this choice, $p_{\boldsymbol{z}'}^{(n)}$ is essentially the mean of all the auxiliary distributions, from $t(n)$ to $t = T-1$. We refer to this variant as *soft simulated annealing* and exemplify it in Fig. 5b.

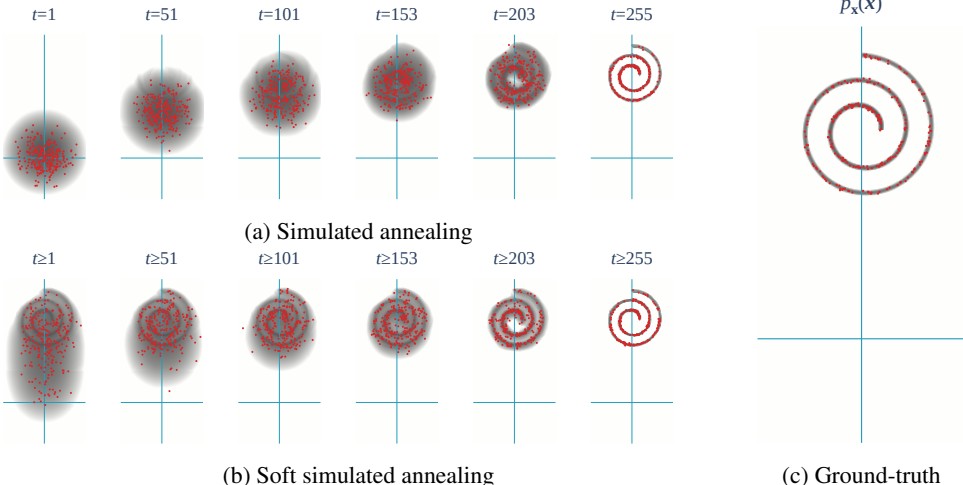

(a) Simulated annealing

(b) Soft simulated annealing        (c) Ground-truth

Figure 5: We depict intermediate samples along with the underlying distribution from the basic simulated annealing (a) and the soft simulated annealing (b) processes applied to the learned spiral toy model. The ground truth distribution is shown in pane (c).

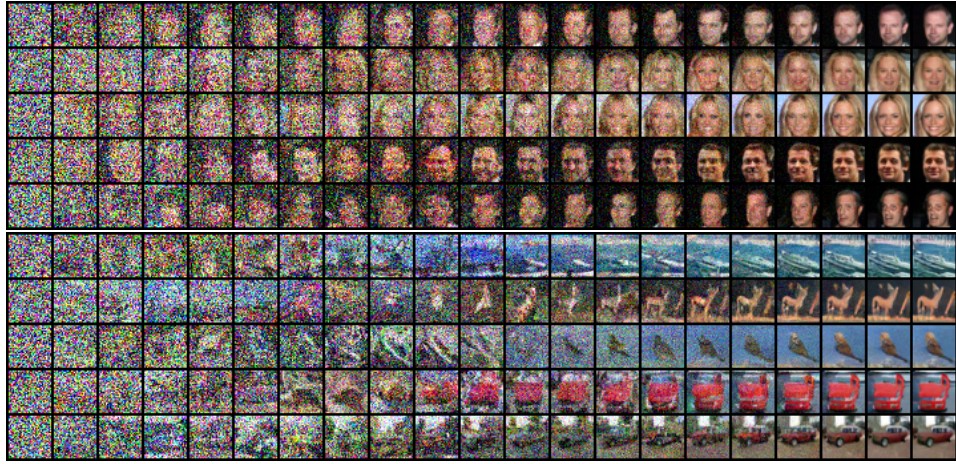

(a) Sample generation using simulated annealing from a model trained with CD+CE.

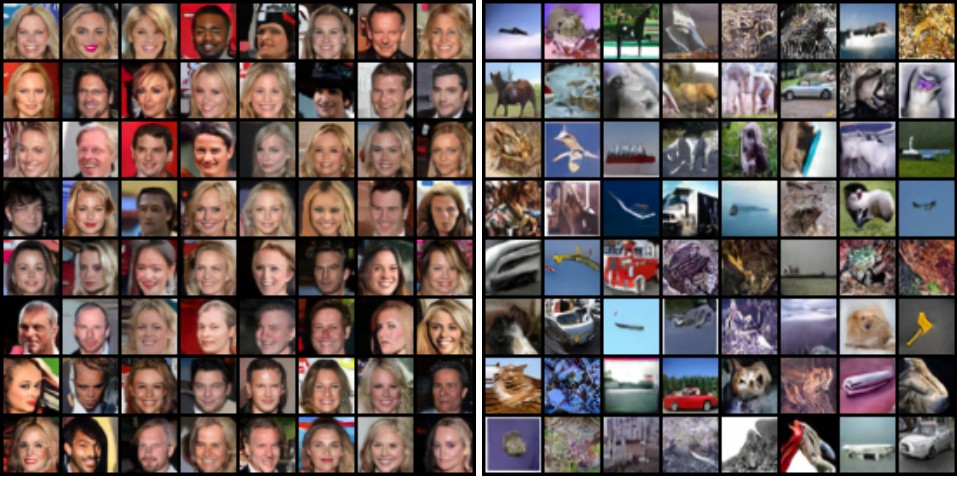

(b) Images generated from models trained on CelebA and CIFAR10 using CD+CE.

Figure 6: Results on CIFAR10 and CelebA.

## 4 EXPERIMENTS

We now illustrate the efficacy of our method on a toy problem as well as on the CFIAR10 and CelebA datasets. Code for all experiments will be released upon acceptance of the paper.

### 4.1 TOY MODEL

The toy problem appearing in Fig. 5c involves data lying on a 2D shifted spiral, where the shifting has been introduced to aid the visualization. To apply our method, we selected a Gaussian distribution centered at the origin as a reference distribution and defined 256 auxiliary distributions, according to (1). We then used JointCD to train a neural network. For the MCMC process, we used Metropolis-Hastings adjusted Langevin dynamics with $K = 3$ steps. The details of the network and the training process can be found in App. C. The resulting learned model is shown in Fig. 5a.

### 4.2 CELEB A & CIFAR10

We trained models using both CD+CE and JointCD on CIFAR10 and on CelebA at $32 \times 32$ resolution. Here we have used 1024 auxiliary distribution, also according to the linear interpolation in (1) As our parametric model, we used ResNet18 (He et al., 2016) with minor changes. Here as well, we used adjusted Langevin dynamics, but during training we gradually increased the lengths of the

Table 1: FID and inception scores for methods trained unconditionally on CIAFR10

| Model | Inception ↑ | FID ↓ |
|---|---|---|
| **GAN-based** | | |
| SNGAN (Miyato et al., 2018) | 8.22 | 21.7 |
| StyleGAN2-ADA (Karras et al., 2020) | 9.83 | 2.92 |
| **Normalizing flows** | | |
| Residual flow (Chen et al., 2019) | | 46.37 |
| FCE (Gao et al., 2020a) | | 37.3 |
| **Score based** | | |
| NCSN-v2 (Song & Ermon, 2020) | 8.4 | 10.87 |
| DDPM (Ho et al., 2020) | 9.46 | 3.17 |
| **EBMs (ML based)** | | |
| CoopNets (Xie et al., 2018) | 6.55 | 33.61 |
| Multi-grid EBM (Gao et al., 2018) | 40.01 | 6.56 |
| CF-EBM (Zhao et al., 2020) | | 16.71 |
| EBM-DRL (Gao et al., 2020b) | 8.3 | 9.58 |
| **EMBs (CD based)** | | |
| EBM-IG (Du & Mordatch, 2019) | 6.78 | 38.2 |
| Improved-EBM (Du et al., 2021) | 7.85 | 25.1 |
| **JointCD (Our)** | 9.09 | 26.8 |
| **CD+CE (Our)** | 8.29 | 23.7 |

chains, from $K = 5$ to $K = 100$. It is worth noting that due to the need to take a large number of MCMC steps before each gradient step, the training is slow and took 7 days on 4 RXT-2080Ti GPUs. The full details of the network and the training process can be found in App. C.

For generating samples, we used simulated annealing. We found that on CIFAR10, the CD+CE method performs better than JointCD. The generation process is visualized in Fig. 6a, and generated samples are shown in Fig. 6b. These results are all from the CD+CE model. Results from the JointCD method can be found in App. D.

We used the inception score (Salimans et al., 2016) and FID (Heusel et al., 2017) to evaluate the CIFAR10 results [3]. With the CD+CE method, we achieved an inception score of 8.5 and an FID score of 23.7. As can be seen in Table 1, these results improve upon previous CD based techniques.

## 5 CONCLUSION

We presented new methods for harnessing coarse-to-fine series of distributions for learning EBMs. Our approach views the "time" index of the series as a random variable and defines an auxiliary task of learning the joint distribution of "time" and samples. We illustrated how using this approach in conjunction with the CD method, leads to substantially improved results. One limitation of our method is that it requires relatively long training times. However, we believe that with further hyper-parameter tuning and correct architectural choices, this can be somewhat alleviated in the future. Our joint modelling approach can in principle be used within other generative models, but we leave those directions for future research.

---

[3] Using the PyTorch implementation from https://pypi.org/project/pytorch-gan-metrics/, which has been shown to reproduce the scores of the original implementation with an error smaller then 0.2%.

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

## A   WHY JOINTCD OBEYS DETAILED BALANCE

A sufficient condition for an MCMC process to converge is that it maintains a relation that is know as detailed balance between the conditional distribution of the transitions and the underlying distribution from which we would like to sample. Specifically given an underlying distribution $p_{\mathbf{z}}(\boldsymbol{z})$ and an MCMC process with a conditional probability of $p_{\tilde{\mathbf{z}}|\mathbf{z}}(\tilde{z}|z)$ to step form $\boldsymbol{z}$ to $\tilde{z}$, then detailed balance criteria is given by:

$$\frac{p_{\tilde{\mathbf{z}}|\mathbf{z}}\left(\tilde{\mathbf{z}}|\mathbf{z}\right)p_{\mathbf{z}}\left(\mathbf{z}\right)}{p_{\tilde{\mathbf{z}}|\mathbf{z}}\left(\mathbf{z}|\tilde{\mathbf{z}}\right)p_{\mathbf{z}}\left(\tilde{\mathbf{z}}\right)}=1 \tag{10}$$

That exist various MCMC process that operate on continuous random variables and obey (usually approximately) the detailed balance criteria. In order to apply one of these existing MCMC process to the joint distribution of $p_{\mathbf{z},\mathrm{t}}$ we suggest the following way to take a step from $(\boldsymbol{z},t)$ to $(\tilde{vz},\tilde{t})$. In each step we begin by ignoring $t$ and generate $\tilde{z}$ using the continuous MCMC process on $\boldsymbol{z}$ with regard to the marginal distribution $p_{\mathbf{z}}$. We then sample $\tilde{t}$ based on $\tilde{z}$ according to the conditional distribution $p_{\mathrm{t}|\mathbf{z}}$. *I.e.,* :

$$p_{\tilde{\mathbf{z}},\tilde{\mathrm{t}}|\mathbf{z},\mathrm{t}}\left(\tilde{\mathbf{z}},\tilde{t}|\mathbf{z},t\right)=p_{\mathrm{t}|\mathbf{z}}\left(\tilde{t}|\tilde{\mathbf{z}}\right)p_{\tilde{\mathbf{z}}|\mathbf{z}}\left(\tilde{\mathbf{z}}|\mathbf{z}\right) \tag{11}$$

As long as the continuous MCMC process obeys the detailed balance criteria than so does the suggested process with respect to $p_{\mathbf{z},\mathrm{t}}$. This can be seen from following derivation:

$$\frac{p_{\tilde{\mathbf{z}},\tilde{\mathrm{t}}|\mathbf{z},\mathrm{t}}\left(\tilde{\mathbf{z}},\tilde{t}|\mathbf{z},t\right)p_{\mathbf{z},\mathrm{t}}\left(\mathbf{z},t\right)}{p_{\tilde{\mathbf{z}},\tilde{\mathrm{t}}|\mathbf{z},\mathrm{t}}\left(\mathbf{z},t|\tilde{\mathbf{z}},\tilde{t}\right)p_{\mathbf{z},\mathrm{t}}\left(\tilde{\mathbf{z}},\tilde{t}\right)}=\frac{p_{\mathrm{t}|\mathbf{z}}\left(\tilde{t}|\tilde{\mathbf{z}}\right)p_{\tilde{\mathbf{z}}|\mathbf{z}}\left(\tilde{\mathbf{z}}|\mathbf{z}\right)p_{\mathbf{z},\mathrm{t}}\left(\mathbf{z},t\right)}{p_{\mathrm{t}|\mathbf{z}}\left(t|\mathbf{z}\right)p_{\tilde{\mathbf{z}}|\mathbf{z}}\left(\mathbf{z}|\tilde{\mathbf{z}}\right)p_{\mathbf{z},\mathrm{t}}\left(\tilde{\mathbf{z}},\tilde{t}\right)} \tag{12}$$

$$=\frac{p_{\mathrm{t}|\mathbf{z}}\left(\tilde{t}|\tilde{\mathbf{z}}\right)p_{\tilde{\mathbf{z}}|\mathbf{z}}\left(\tilde{\mathbf{z}}|\mathbf{z}\right)p_{\mathrm{t}|\mathbf{z}}\left(t|\mathbf{z}\right)p_{\mathbf{z}}\left(\mathbf{z}\right)}{p_{\mathrm{t}|\mathbf{z}}\left(t|\mathbf{z}\right)p_{\tilde{\mathbf{z}}|\mathbf{z}}\left(\mathbf{z}|\tilde{\mathbf{z}}\right)p_{\mathrm{t}|\mathbf{z}}\left(\tilde{t}|\tilde{\mathbf{z}}\right)p_{\mathbf{z}}\left(\tilde{\mathbf{z}}\right)} \tag{13}$$

$$=\frac{p_{\tilde{\mathbf{z}}|\mathbf{z}}\left(\tilde{\mathbf{z}}|\mathbf{z}\right)p_{\mathbf{z}}\left(\mathbf{z}\right)}{p_{\tilde{\mathbf{z}}|\mathbf{z}}\left(\mathbf{z}|\tilde{\mathbf{z}}\right)p_{\mathbf{z}}\left(\tilde{\mathbf{z}}\right)} \tag{14}$$

$$=1. \tag{15}$$

## B   THE SAMPLING ALGORITHMS

We outline in alg. 3 the process for generating samples from the model using the common simulated annealing, and in alg. 4 the process of using the generalized simuulated annealing described in section 3.5.

| **Algorithm 3:** Basic Simulated Annealing | **Algorithm 4:** General Simulated Annealing |
|---|---|
| Draw a sample $\mathbf{z}$ from $p_{\mathbf{n}}=p_{\mathbf{z}|\mathrm{t}=0}$ | Draw a sample $\mathbf{z}$ from $p_{\mathbf{n}}=p_{\mathbf{z}|\mathrm{t}=0}$ |
| **for** $t=0$ *to* $T-1$ **do** | **for** $n=0$ *to* $N$ **do** |
| $\quad$ **for** $n=0$ *to* $N$ **do** | $\quad \hat{p}_{\mathbf{z}';\boldsymbol{\theta}}^{(n)}:=\sum_{t=0}^{T-1}\hat{p}_{\mathbf{z};\boldsymbol{\theta}}(\boldsymbol{z},t)\frac{p_{\mathrm{t}}^{(n)}(t)}{p_{\mathrm{t}}(t)}$ |
| $\quad\quad \tilde{z}\leftarrow\mathcal{T}_{\hat{p}_{\mathbf{z}|\mathrm{t}=t;\boldsymbol{\theta}}}(\tilde{z})$ | $\quad \tilde{z}\leftarrow\mathcal{T}_{\hat{p}_{\mathbf{z}';\boldsymbol{\theta}}^{(n)}}(\tilde{z})$ |
| $\quad$ **end** | **end** |
| **end** | |

## C   EXPERIMENT DETAILS

### C.1   TOY MODEL

For the toy model, we used a network of 4 residual blocks containing fully connected layers with a width of 256. As a reference distribution we used white Gaussian distribution with zero mean and STD $\sigma_{\mathbf{n}}=0.3$. We found it beneficial to have the STD of the reference distribution slightly larger then that of the data. We trained the model using JointCD (Alg. 2) with Metropolis-Hastings adjusted Langevin dynamics. The Langevin step size was been adaptively adjusted during the run to maintain an average acceptance rate of 60% in the Metropolis-Hastings adjustment stage. This was done by keeping an array of an individual step size for each value of $t$. We have found the step size of each $t$ to converge in an early state of the training to about $0.4\beta_t\sigma_{\mathbf{n}}$, where

We have used Adam (Kingma Diederik & Adam, 2014) as an optimizer with a learning rate of $10^{-3}$ and the default momentum values of $\beta_1 = 0.9, \beta_2 = 0.999$. We used a batch size of 256. The model was trained for 400 epochs (30 minutes on a RTX 2080Ti GPU).

## C.2  CIFAR10 AND CELEBA 32x32

As our model we have used ResNet18 with the following modifications:

1. We have removed the BatchNorm (Ioffe & Szegedy, 2015) layers which, makes the network nondeterministic par input.
2. We have replaced the ReLU activation with LeakyReLU.
3. We have added a cumulative-summation layer before the output of the network as described in section 3.2.

As a reference distribution, we have again used a white Gaussian distribution with zero mean and STD of $\sigma_{\mathbf{n}} = 0.5$ and we have defined a series of $1024$ distribution according to (1). When working with the images we have subtracted a value of $0.5$ from all the pixels in order to roughly center the data around $0$.

We have used the same MCMC process as in the toy model along with the same method for adjusting the step size. We have used AdamW (Loshchilov & Hutter, 2018) as an optimizer with a weight decay of $10^{-3}$. We have used an initial learning rate of $10^{-3}$ with a linear warm-up during the first 4000 steps. We have reduced the learning rate by a factor of 10 every 200,000 steps over the full training process which consisted of 300,000 steps. We used a batch size of 512.

The number of steps Langevin dynamic has been set to 5 during the first 40000 and was linearly increased up to 100 over the course of the next 40000 steps. From there on the number of steps has been fixed to 100. Due to this large number of Langevin steps, the training time was relatively long and took 7 days on 4 RTX 2080Ti GPUs.

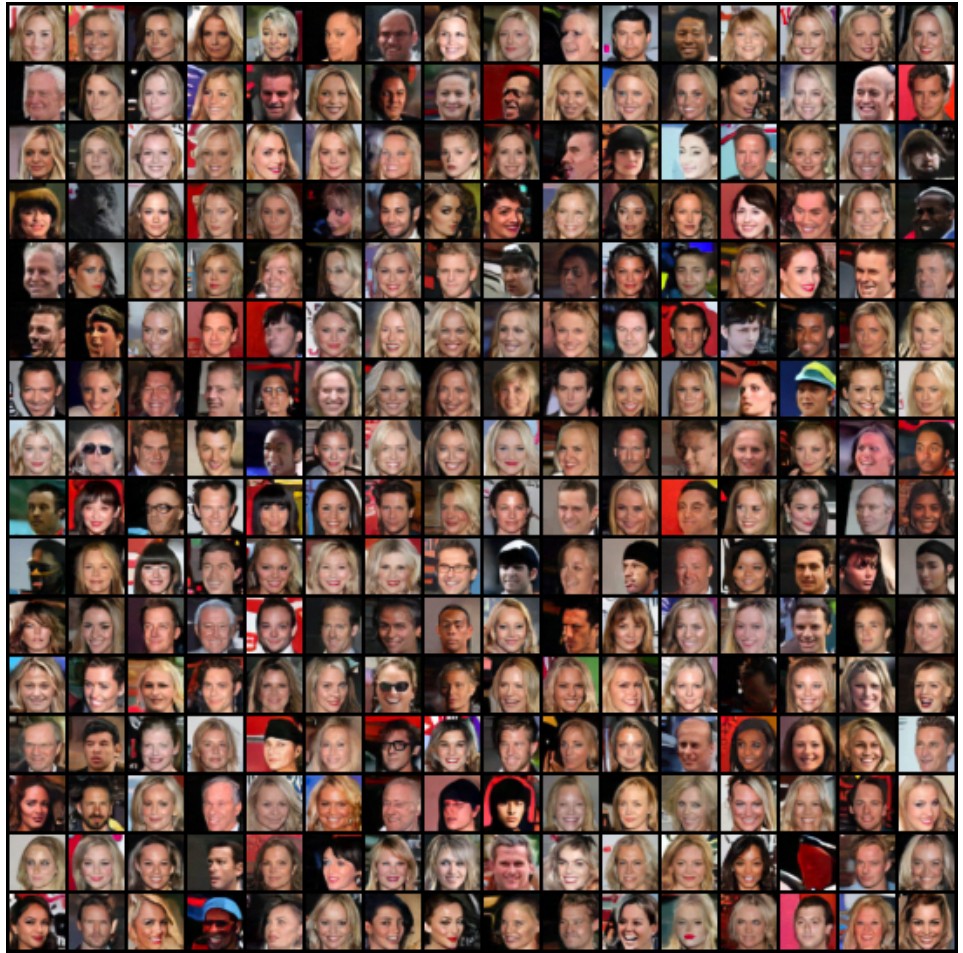

Figure 7: Generated images from CelebA using a model trained using CD+CE

# D  ADDITIONAL RESULTS

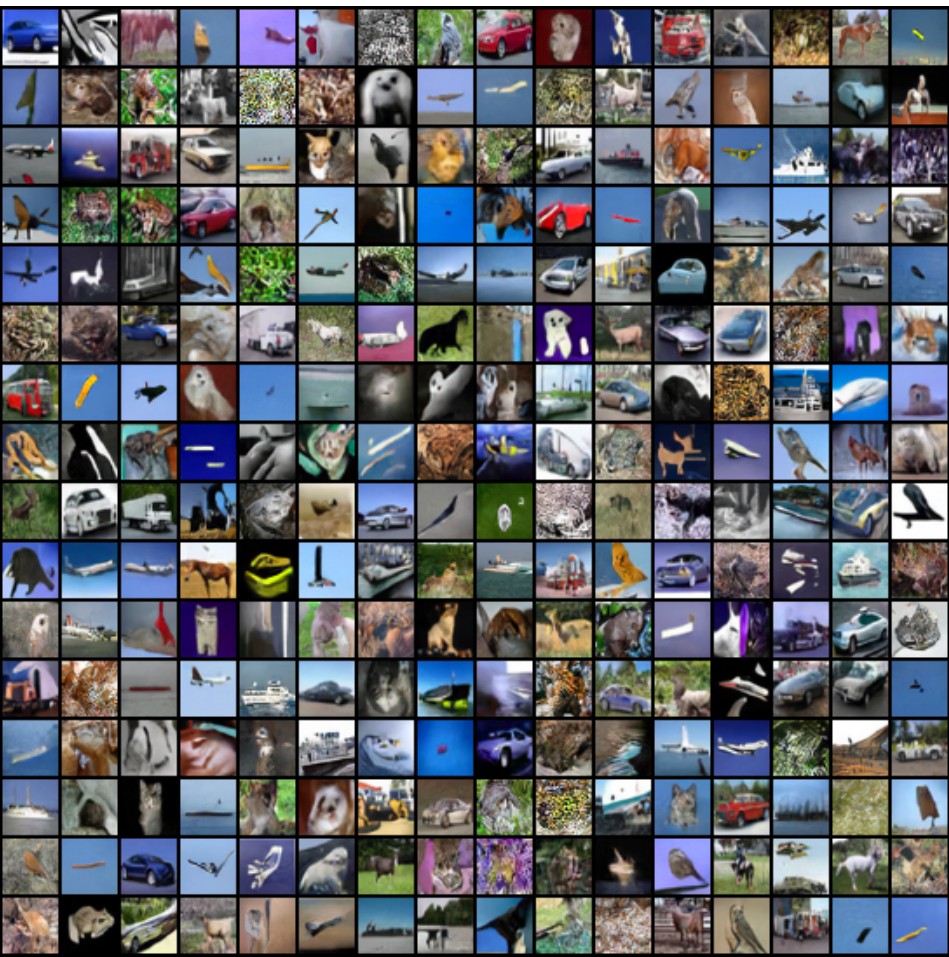

Figure 8: Generated images from CIFAR10 using a model trained using CD+CE

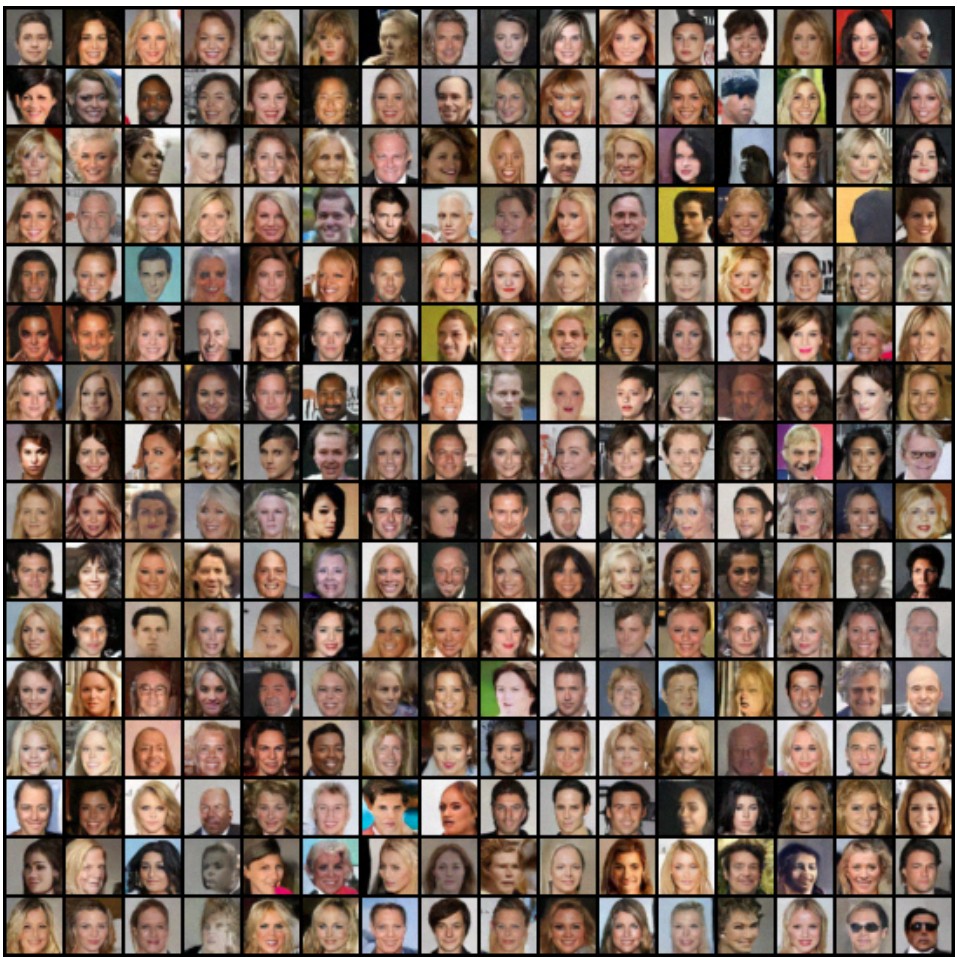

Figure 9: Generated images from CelebA using a model trained using JointCD

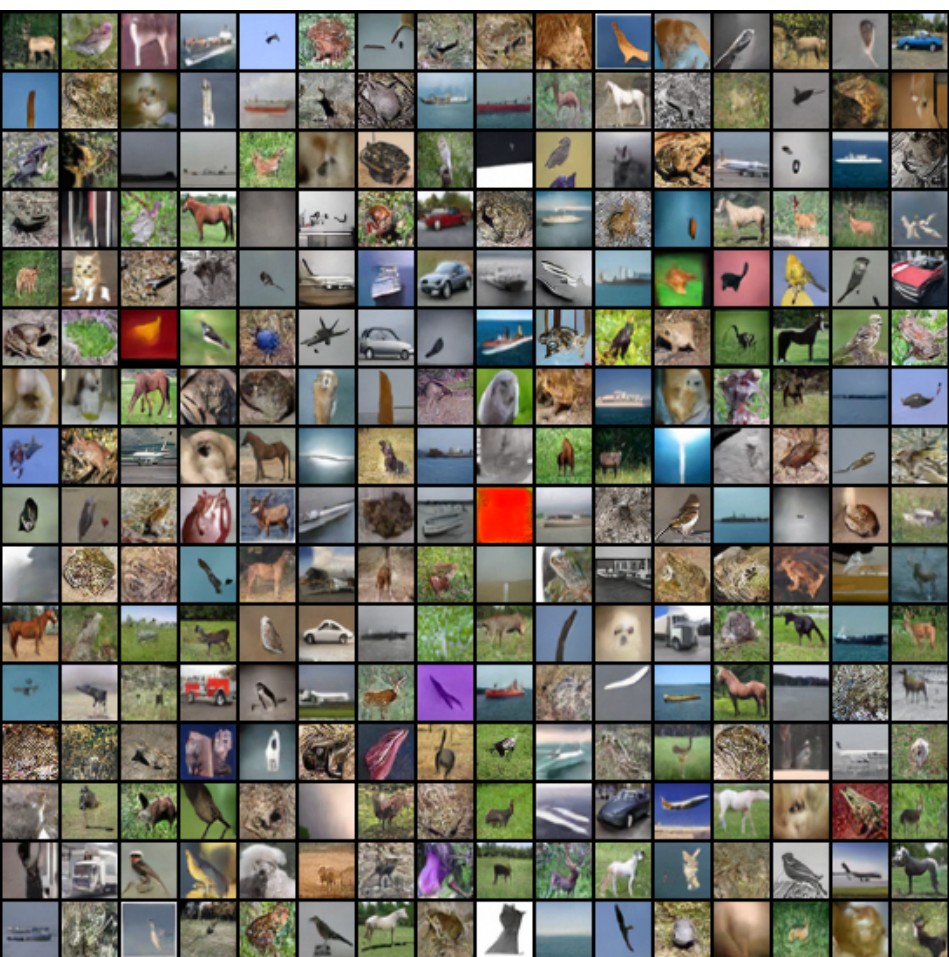

Figure 10: Generated images from CIFAR10 using a model trained using JointCD

