# OpenReview forum: "Thinking fourth dimensionally: Treating Time as a Random Variable in EBMs"
_ICLR.cc/2023/Conference — Submitted to ICLR 2023_

### Official Review · Reviewer_KRWa · 2022-10-24

**Confidence:** 3
**Correctness:** 2
**Technical Novelty And Significance:** 2
**Empirical Novelty And Significance:** 2
**Recommendation:** 5

**Clarity, Quality, Novelty And Reproducibility:**

* Clearly written.
* An interesting take on diffusion-/score-based modelling.

**Strength And Weaknesses:**

**Strengths**
* The paper proposes an interesting view on diffusion and score-based models that leads to a different formulation and training criteria.
* The paper is well-written and clearly motivated.
* Empirical evaluation appears to be sound.

**Weaknesses**
* The motivation of lack of parameter sharing between models of $p_z(z|t)$ is not convincing. For example, this does not seem to take into account the existence of continuous time diffusion- and score-based models, or the fact that in practice $p_z(z|t)$ are often implemented with neural networks that share most of the parameters.
* Computation efficiency of the proposed model is not discussed. For example, taking K (which runs up to 100) steps of MCMC in each training iteration seems rather expensive, especially when comparing this to diffusion-based models that can generate a sample from an intermediate distribution required for training in a single step. This aspect of the model should be discussed.
* Despite clear motivation the benefits of the proposed method are not obvious form the empirical evaluation. For example, the proposed model has one of the worst FID scores among the competing methods and is far behind the SOTA methods in terms of Inception and FID scores. Why is that?

**Summary Of The Paper:**

The paper presents a generative model that draws on the diffusion-based and score-based models that generate samples through a sequence of intermediate distributions, but proposes different parameterisation, training criteria and sampling procedure. The model is motivated largely by the fact that prior models treat time (that parameterises intermediate distributions) as discrete and do not benefit from parameter sharing between these distributions. Limited model evaluation is presented.

**Summary Of The Review:**

Interesting well-motivated generative model that needs stronger empirical results.

---

> ### Author Response · Authors · 2022-11-18
> **Reply**
>
> Thank you for your insights and comments.
>
> ### Parameter sharing and continuous time
>
> Indeed most methods that model a series of distributions do so use a single conditional network and therefore share most of the parameters between the different models. This creates an inductive bias which should theoretically push the different conditional models to behave in a similar manner. However, in practice, this does not always happen. For example, in annealed score-matching, each model is only correct around a restricted region in the sample space. Our claim is, that by using our method, we can not only have the different models share the majority of the parameters but, in fact, have all the different models trained using samples from all of the different distributions.
>
> ### Computational efficiency
>
> We agree that the long computational time is a major drawback of this method. However, it is not larger than e.g. that of EBM-DRL. In any case, note we mentioned this both in the experiments section and in the conclusion section.
>
> ### Quality of results
>
> First, we would like to point out, that we have been able to improve our JointCD method by slightly tuning hyper-parameters (the new hyper-parameters appear in the updated version). The new Inception score is 9.09 and the new FID score is 26.83. The results are still behind the EBM-DRL method. However, we do believe that the basic concept we presented may be combined with other improvements, like architectural changes, augmentations, better optimization methods, etc. Here, we mainly wanted to focus on the formalism, and left those potential improvements for future work.

---

### Official Review · Reviewer_k6Mq · 2022-10-24

**Confidence:** 4
**Correctness:** 2
**Technical Novelty And Significance:** 2
**Empirical Novelty And Significance:** 2
**Recommendation:** 3

**Clarity, Quality, Novelty And Reproducibility:**

* The paper is written reasonably well but some sections are unclear (see weaknesses).
* Please refer to weaknesses for the evaluation of quality.
* The specific method proposed in this work is novel but its significance is limited.
* The proposed method, as described in the paper, doesn't appear straightforward to implement. In the absence of specific details and/or code, the reproducibility of this work is low.

**Strength And Weaknesses:**

[Strengths]
* I found the idea of modeling a joint distribution by making the neural network output log probabilities for every time index interesting.

[Weaknesses]

Motivation/Claims/Significance:
* The proposed approach of treating time as a random variable is not well-motivated in the introduction. It is unclear why such a treatment is needed or even useful. Furthermore, it is unclear what treating the time as random variable even means in the context of energy-based models. Sampling methods, such as Langevin Monte Carlo, follow the steepest descent curve in the Wasserstein space (given by the Fokker-Plank equation) to draw from the Boltzmann distribution. This curve of probability densities can be indexed by time but I am not sure what treating time as a random variable would imply.
* The only potential benefit that's mentioned in the introduction is that it leads to "improved results". However, the improvement is over a specific class of EBM and is marginal.
* It is claimed that the method can be used "in conjunction with almost any explicit distribution learning algorithm"; however, only a specific variant has been studied in the paper. It is unclear how it would be applicable to other algorithms/models and how that would help.

Writeup:
* Some things in the paper are unclear:
    * Is Eq. (1) just an example that motivates the discussion or is it the form that's used in the experiments. I am not sure if previous works have studied intermediate distributions that are linear combinations in the data space.
    * What is meant by "which the standard approach"? As per my understanding, EBMs learn energy of the Boltzmann distribution and are not indexed or conditioned on time.
    * Why did the authors choose to initialize the MCMC process "with a sample from the dataset"? Can the authors explain what they meant by "vanilla contrastive divergence"?

Technical Contribution and Empirical Evaluation:
* The overall technical contribution of this work is limited and it doesn't offer any new insights.
* The sampling procedure now has an inner loop which appears to increase the computational complexity.
* The authors state that the method results in "substantially improved results". However, scores have been reported only for one dataset and the improvement is marginal, that too over the specific class of CD-based EBMs. CelebA scores are conspicuously missing.

**Summary Of The Paper:**

The paper proposes to treat "time" in an energy-based model as a random variable and learns a joint distribution of time and samples. Concretely, this results in a parameterized function that outputs the joint log probability of the samples and time indices. Two methods of training such a model have been presented: a) using product of conditional and marginal where training is achieved by the optimization of the contrastive divergence and cross entropy losses; b) using the joint distribution directly. The proposed model offers slight improvements over baseline models trained via constrastive divergence.

**Summary Of The Review:**

The key proposal of treating time as a random variable lacks clarity and motivation. The overall technical contribution and significance of this work is limited.

---

> ### Author Response · Authors · 2022-11-18
> **Reply**
>
> Thank you for your insights and comments.
>
> ### The meaning of time as a random variable
>
> Please note that we denoted by "time" the index which runs over the series of distributions (which gradually interpolates between the data's distribution and some well-known one) and not the index of the step within the MCMC process. Using the name "time" mainly follows the notation in recent diffusion model papers in which the time describes the flow of samples through the series of distributions.
>
> ### Improvement over previous methods
>
> We indeed improve upon "vanilla" CD, which is unable to model complex distributions on its own. With our method, the same algorithm is able to achieve results close to the state-of-the-art EBMs. We agree that the gap to the current state-of-of-he-art results achieved by diffusion models remains significant.
>
> ### Applicability to other EBM training methods
>
> The framework described in Secs. 3.1 and 3.2 does not rely on a specific learning method and can be used with any technique for learning EBMs. All we are claiming is that any chosen EBM can be used to learn either the marginal distribution $p_{z}$ or learn the conditional distribution $p_{z}$ along with a classifier that learns $p_{t|z}$.
>
> ### Is it common practice to use a linear combination between training samples and noise?
>
> The linear interpolation in Eq. (1) is indeed the common way in which intermediate samples are generated in the most recent state-of-the-art methods (diffusion models, score matching, diffusion recovery, etc.). We indeed used the linear combination of Eq. (1) in our experiments. Kindly note that this was mentioned in Sec. 4 ("... and defined 256 auxiliary distributions, according to (1)"). We tried to clarify it better in the updated version.
>
> ### Initializing the MCMC with a sample from the dataset
>
> We used the name "vanilla" CD to describe CD in its original form without any changes as it has been originally described in [Hinton, 2002](https://www.cs.toronto.edu/~hinton/absps/tr00-004.pdf). In its original form, the MCMC is initialized using samples from the dataset. From a theoretical point of view, this is significant for ensuring the convergence of the model to the target distribution when using a finite number of steps in the MCMC process, as discussed in [Nijkamp et al. 2019](https://arxiv.org/abs/1904.09770.pdf) and [Yair & Michaeli, 2021](https://arxiv.org/abs/2012.03295).
>
> ### Additional inner loop
>
> We are uncertain as to what additional inner loop you refer to. Assuming that you meant the MCMC process, then this is not an additional loop that we have introduced. This loop appears in all of the previous MCMC-based algorithms in general and in the CD algorithms in particular.
>
> ### The FID and inception scores for CelebA are "conspicuously" missing
>
> These scores are missing since we have only evaluated our method on 32x32 images from CelebA. As such, we had no reference scores to compare to. First, most papers publish FID scores and trained models for CelebA images starting from a 64x64 resolution. Second, it is meaningless to use Inception scores on CelebA as this score is based on classification of Imagenet categories.

---

> > ### Comment · Reviewer_k6Mq · 2022-11-23
> > **Response**
> >
> > Thanks a lot for your response. Overall, I am still of the opinion that the proposed method lacks motivation (and clarity). The empirical contributions and discussions are not particularly strong either. Hence, I am keeping my score.

---

### Official Review · Reviewer_qaem · 2022-10-25

**Confidence:** 4
**Correctness:** 4
**Technical Novelty And Significance:** 2
**Empirical Novelty And Significance:** 2
**Recommendation:** 5

**Clarity, Quality, Novelty And Reproducibility:**

The core idea is novel and interesting and the presentation in clear. Experiments show a modest improvement over some recent EBMs but the quality of the experimental results is average.

**Strength And Weaknesses:**

STRENGTHS:

1. The technical innovations in this work are an interesting application of design principles in diffusion models to EBM learning. The CD+CE method is an elegant way of training the joint density that relates nicely to existing work.

WEAKNESSES:

1. I am not sure that the claims about the method being the first successful CD-based method are valid (paragraph right before Section 2). The Multigrid method (Gao et al. 2018) and Diffusion Recovery (Gao et al. 2020) use transformed data samples for both initial MCMC samples and for the positive samples. One could make the argument that these works use CD if we broaden the definition of CD to include samples that can be created by transforming the data, as done in the present work.
2. The empirical results in terms of FID score are solid for EBMs but not exceptional. I have some doubt about the reported Inception scores of 8.4 and 8.5, which are much higher than the Inception scores of models with much better FID. Although I realize that Inception and FID can be at odds, this is not generally the case on simple datasets like CIFAR-10. Further experiments at a higher resolution beyond 32x32 or with more diverse datasets would improve the experimental evaluation.
3. EBM training times are already a critical issue, and further increases in runtime are a significant drawback.


**Summary Of The Paper:**

This paper proposes to train EBMs that can model the distribution defined by uniformly sampling a linear interpolation, called time, between a noise distribution and the data distribution. The joint distribution of the interpolated image distribution and the time random variable is the target density, and samples from the target density can be obtained by jointly sampling from noise, data, and the interpolation proportion. Two methods are proposed to learn the joint distribution. The first uses joint MCMC sampling to obtain the negative samples and standard CD loss. The second uses a JEM-like objective where the EBM parameterizes the marginal of the image distribution via a log sum exp over the output of the EBM output, which is a vector of the conditional densities for each class. One can draw samples of the model by initializing states from noise and performing a few Langevin updates on at each level of time until the time step corresponding to the data distribution is reached. Experiments evaluating the method are performed on CIFAR-10 and Celeb-A.

**Summary Of The Review:**

Overall, I found this work interesting but not exceptional. The core idea of the proposed method is the biggest strength of the paper, and it could prompt further design of diffusion-like sequences of distributions from a reference distribution to the data. The experimental results are reasonable but not extremely compelling and the gap between EBMs and diffusion models remains wide. The high computational cost could be an obstacle towards experiments with more complex datasets. I am on the borderline but inclined to recommend against accepting.

---

> ### Author Response · Authors · 2022-11-18
> **Reply**
>
> Thank you for your insights and comments.
>
> ### The first to successfully apply "vanilla" CD to image distributions
>
> In our paper, we distinguish between using the CD in its original ("vanilla") form, in which the MCMC is initialized using samples from the dataset, and alternative methods, which initialized the MCMC using samples from a preselected distribution (or a degraded version the samples). The second method is basically an approximated maximum likelihood, in which the partition function of the EBM is approximated using samples from the distribution of the learned model.
> While the two methods are very similar in their implementation, they rely on significantly different theoretical justifications. The "vanilla" CD indeed suffers from a range of difficulties, but it does not assume that the MCMC process has converged to the distribution of the model. On the other hand, the approx. ML method does, in theory, rely on the convergence of the MCMC.
> To the best of our knowledge, under this distinction, "vanilla" CD has yet to be successfully used for modeling complex distributions such as those of images.
>
> ### The differences between the FID and inception score
>
> We agree that the inception score is a very problematic measurement of visual quality (as discussed in [Barratt & Sharma, 2018](https://arxiv.org/abs/1801.01973.pdf)). The scores in the paper have been calculated using the [pytorch-gan-metrics](https://pypi.org/project/pytorch-gan-metrics/) package, which has been shown to produce results similar to does of the original implementation up to an error of $<0.2\%$. We intend to share our code and trained networks that have produced these results.
>
> ### Training and running times
>
> We agree that the training and the sample generation times are a main drawback. However, we believe that with further tuning, these can be reduced, similarly to the evolution we have seen in recent years in other types of generative models. Here, we wanted to put more focus on the formalism rather than on the implementation.

---

### Decision · Program_Chairs · 2023-01-20

**Decision:**

Reject

**Justification For Why Not Higher Score:**

Please see weaknesses listed above:

Weaknesses: Unanimously there were three concerns with the paper:
(a) The technical formulation - treating time as a random variable - was not particularly strongly motivated.
(b) EBM training times are already a critical issue, and further increases in runtime are a significant drawback.
(c)  The empirical results - on limited datasets - in terms of improvements over baselines are not convincing.


**Justification For Why Not Lower Score:**

N/A.

**Metareview: Summary, Strengths And Weaknesses:**

Summary: This paper proposes blends a coarse-to-fine noise scale strategy integral to training diffusion models, with EBM training where the noise scale - treated as a time dimension - is treated as a random variable and jointly modeled along with the data.

Strengths: Generally, the reviewers found the core ideas to be an interesting variant of EBMs, with clear presentation and sound empirical analyses (although not entirely convincing results).

Weaknesses: Unanimously there were three concerns with the paper:
(a) The technical formulation - treating time as a random variable - was not particularly strongly motivated.
(b) EBM training times are already a critical issue, and further increases in runtime are a significant drawback.
(c)  The empirical results - on limited datasets - in terms of improvements over baselines are not convincing.